



# A new L4 multi-sensor ice surface temperature product for the Greenland Ice Sheet

Ioanna Karagali[1], Magnus Barfod Suhr[1], Ruth Mottram[1], Pia Nielsen-Englyst[1,2], Gorm Dybkjær[1], Darren Ghent[3], and Jacob L. Høyer[1]

[1]Danish Meteorological Institute, Lyngbyvej 100, Copenhagen-Ø, DK-2100, Denmark
[2]DTU-Space, Technical University of Denmark, Lyngby, DK-2800, Denmark
[3]National Centre for Earth Observation (NCEO), University of Leicester, University Road, Leicester, LE1 7RH, UK.

**Correspondence:** Ioanna Karagali (ika@dmi.dk)

**Abstract.**

The Greenland Ice Sheet (GIS) is subject to amplified impacts of climate change and its monitoring is essential for understanding and improving scenarios of future climate conditions. Surface temperature over the GIS is an important variable as it regulates processes related to the exchange of energy and water between the surface and the atmosphere. As few key local

observation sites exist, an important alternative to obtain surface temperature observations over the GIS is space-borne sensors that carry thermal infrared instruments. These offer several passes per day with a wide view and are the basis of deriving Ice Surface Temperature (IST) products.

The aim of this study was to compare several satellite IST products for the GIS and develop and validate the first multi-sensor, gap-free (Level 4, L4) product for 2012. High resolution Level 2 (L2) IST products from the European Space Agency

(ESA) Land Surface Temperature Climate Change Initiative (LST_cci) project and the Arctic & Antarctic Ice Surface Temperatures from Thermal Infrared Satellite Sensors (AASTI) dataset, were assessed using observations from the PROMICE stations and IceBridge flight campaigns. The AASTI data showed overall better performance compared to LST_cci data, that in return had superior spatial coverage and availability. Both datasets were further utilised to construct a daily, gap-free, L4 IST product using the optimal interpolation (OI) method. The resulting L4 IST product performed satisfactorily in terms of

quality when compared with surface temperature observations from the PROMICE stations and IceBridge flight campaigns. By combining the advantages of the upstream satellite datasets, the gap-free L4 IST product allowed for the analysis of IST over the GIS during the year 2012, when a significant melt event occurred. Mean summer (June-August) IST over the GIS was −5.5°C±4.5°C, with an annual mean of −22.1°C±5.4°C. Mean IST during the melt season (May-August) ranged from −15°C to −1°C, while almost the entire GIS experienced at least between 1 and 5 melt days when temperatures were −1°C or higher.

Finally, this study assessed the potential for using the satellite L4 IST product to improve model simulations of the GIS surface mass budget (SMB). The new L4 IST product was first used to evaluate, and then it was assimilated into an SMB model of snow and firn processes for the year 2012, when extreme melting occured, to assess the impact of including a high resolution IST product on the SMB model. Compared with independent observations from the PROMICE stations and IceBridge flight campaigns, inclusion of the L4 IST dataset improved the model performance during the key onset of the melt



season, where model biases are typically large. Reassuringly, internal model parameterisations performed well compared to the L4 IST dataset outside of this season, providing more confidence in modelled GIS surface mass budget (SMB) estimates.

# 1  Introduction

There has been an increase in the rate of Arctic ice loss over the last two decades, including accelerated retreat of glaciers and ice sheets, a reduction in the extent and timing of seasonal snow and a reduced sea ice extent and thickness as a result of climate
change (IPCC, 2019). The Greenland Ice Sheet (GIS) has had a net negative mass balance for at least the last 25 years, resulting from the combination of increased dynamic thinning, i.e. loss due to accelerated flow and calving, and decreased surface mass balance (SMB) (Shepherd et al., 2020; Mottram et al., 2019; Mankoff et al., 2020). From September 2019 to August 2020 the GIS experienced ice loss higher than the average for the period from 1981 to 2010 (Moon et al., 2020).

SMB is the budget of accumulation, via snowfall, and ablation via melt and runoff. Also important are processes of meltwater
percolation into the snow and firn (snow that has survived at least one annual cycle) where it may survive as liquid or refreeze forming large ice layers (Broeke et al., 2009; Ettema et al., 2010; Machguth et al., 2016; Reijmer et al., 2012). A decrease in SMB over the last 25 years is the largest contributor to the mass loss of the GIS, largely due to enhanced melting during the summer melt season (Shepherd et al., 2020). SMB is directly measured at only a few point locations in Greenland and regional climate models (RCMs) are therefore used to make integrated estimates over the whole ice sheet, e.g. Shepherd et al.
(2020). However, as analysis by Fettweis et al. (2020) shows there are large discrepancies between models in terms of both the components of SMB (mainly melt rates and precipitation) and the geographical pattern of SMB. Satellite observations over large areas are therefore essential for evaluating models and are also useful in process studies, to identify sources of model error as well as being potentially useful in correcting model biases via assimilation in climate and weather models.

Land surface temperature (LST) can be observed by satellites and is classified as an Essential Climate Variable (ECV)
according to the Global Observing System for Climate (GCOS) (GCOS, 2016). The temperature of the snow and ice covered land surfaces (Ice Surface Temperature, IST), usually calculated from the surface energy balance from observations or RCMs, controls both melt rates and other snow pack processes that are important for characterising SMB. As firn provides an important buffer for meltwater, it must be included in SMB models to determine rates of ice sheet loss. However there is a wide variation in the performance of different firn models and the parameterisations used within them (Langen et al., 2017; Vernon et al.,
2013). As an example, retention and refreezing of liquid water are modulated by grain size and densification, which are in turn strongly influenced by surface temperatures over the GIS (Vandecrux et al., 2020). Improving modelled IST is therefore important for SMB estimates of the GIS.

Satellite observations in the thermal infrared (IR) have allowed monitoring of clear-sky ISTs over the last four decades. Infra-red sensors operate in the atmospheric window where wavelengths range between $10\mu m$ and $12\mu m$, thus measurements refer
to the skin Ice Surface Temperature, which may differ considerably from, e.g. in-situ stations which typically measure the 2-m air temperature. Furthermore, infra-red measurements can only be obtained under clear-sky conditions, thus are representative of instances when temperature is lower compared to cloudy conditions (Nielsen-Englyst et al., 2019).



The primary IR sensors used for estimating IST are the Advanced Very High Resolution Radiometer (AVHRR) available since August 1981 (Comiso and Hall, 2014; Dybkjær et al., 2012; Comiso, 2003), the Moderate-Resolution Imaging Spectro-radiometer (MODIS) available since 1999 (Hall et al., 2008, 2012, 2013), the (Advanced) Along Track Scanning Radiometers, (A)ATSRs, available since 1991 (Dodd et al., 2019) and the Sea and Land Surface Temperature Radiometer (SLSTR) A/B as successors of the (A)ATSR series. Recently, the European Space Agency (ESA) funded LST_cci project, part of the Agency's Climate Change Initiative (CCI) Programme, released initial versions of data products from several satellites that provide temperature information across land surfaces regionally and globally, with a temporal extent of 20 years (Perry et al., 2020). More recently, Nielsen-Englyst et al. (2021) used clear-sky skin temperature observations from space-bourne infra-red radiometers to derive daily mean clear-sky 2 m air temperatures ($T_{2m}$) in the Arctic, including the Greenland Ice Sheet.

Averaged clear-sky IST observations from AVHRR have previously been analyzed and used to calculate climate trends in the Arctic (Comiso, 2003; Wang and Key, 2005). However, the clear-sky limitation of IR observations usually results in differences when compared to the averaged ISTs measured during all-sky conditions (Comiso, 2003; Koenig and Hall, 2010; Nielsen-Englyst et al., 2019), which may impact the accuracy of the observed trends (Liu et al., 2008). Currently, no gap-free (level 4, L4) IST product exists for the GIS.

This study presents the results from a user case study within the ESA LST_cci project about the uptake of the first satellite multi-sensor, optimal-interpolated L4 IST fields covering the GIS during 2012. IST data from IR satellite sensors were used from the ESA LST_cci project as well as from the Arctic and Antarctic Ice Surface Temperatures from thermal Infrared satellite sensors (AASTI) data set (Dybkjaer et al., 2018; Høyer et al., 2019). The individual satellite products of IST were inter-compared and validated against in situ measured radiometric surface temperatures from the PROMICE stations and IceBridge flight campaigns. The year 2012 was selected due to extensive surface melt over most of the ice sheet (Box et al., 2012; Nghiem et al., 2012; Hall et al., 2013) resulting in the lowest SMB on record, and a challenging event for climate models, many of which underestimated the contribution to melting of turbulent heat fluxes, especially the sensible heat flux (Fausto et al., 2016). Finally, this study assessed the potential to integrate the L4 IST product into climate models by first evaluating against and then assimilating the daily L4 IST data into a SMB model forced by the RCM HIRHAM5 (Langen et al., 2015, 2017).

## 2   Data

### 2.1   Satellite Data

As the GIS is located at high latitudes, it is not feasible to use aggregated day/night observations from ascending and descending orbits. L2 v1.0 data (without the subsequent enhancements to the algorithms, cloud masking and uncertainties) from the ESA LST_cci project were used, consisting of 1-km IST observations from the AATSR on ENVISAT (only available until April 2012) and the MODIS on the Aqua and Terra platforms (LST_cci, 2020). In addition, IST observations from the AASTI v2 L2 data set, available multiple times per day with 4 km resolution, were included in the analysis (Dybkjaer et al., 2018; Høyer et al., 2019). This data set is generated using Global Area Coverage (GAC) retrievals from the AVHRR onboard the NOAA



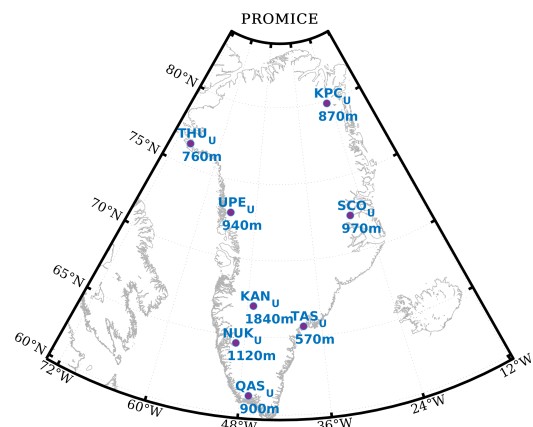

**Figure 1.** Locations of the used PROMICE stations along the Greenland Ice Sheet and their elevation.

and METOP platforms. The AASTI dataset extends north of 50°N and south of 50°S, providing surface temperatures for sea ice, land ice, open water and marginal ice zone areas from 1982 to 2015. The satellite products assessed in this study are listed in Table 1.

**Table 1.** Input satellite data used for the derivation of the daily L4 IST.

| Product String | Version | Sensor type | Resolution | Data availability |
|---|---|---|---|---|
| ENVISAT AATSR L2P | 1.0 | IR | 1 km swath | Jan-Apr, 2012 |
| TERRA MODIS L2P | 1.0 | IR | 1 km swath | Jan-Dec, 2012 |
| AQUA MODIS L2P | 1.0 | IR | 1 km swath | Jan-Dec, 2012 |
| AASTI AVHRR GAC | 2.0 | IR | 4 km swath | Jan-Dec, 2012 |

## 2.2 PROMICE

The Programme for Monitoring of the Greenland Ice Sheet (PROMICE) data are provided by the Geological Survey of Denmark and Greenland (Ahlstrøm et al., 2008; van As and Fausto, 2011; Fausto et al., 2021). The surface temperatures are derived from up-welling longwave radiation measured by Kipp and Zonen CNR1 or CNR4 radiometers by assuming an emissivity of 0.97 (Fausto et al., 2021). Only PROMICE data from the upper ablation zone and accumulation zone were used to ensure that data are only acquired over permanently snow- or ice- covered surfaces. Figure 1 shows the geographical distribution of the
eight selected PROMICE stations and their elevation, also listed in Table 2.





**Table 2.** PROMICE observation sites including the surface type, co-ordinates and elevation.

| Site | Station | Surface Type | Latitude °N | Longitude °W | Elevation (m) |
|---|---|---|---|---|---|
| Kangerlussuaq | KAN_U | ACC | 67.00 | 47.03 | 1840 |
| Crown Prince Christian Land | KPC_U | ACC | 79.83 | 25.17 | 870 |
| Scoresbysund | SCO_U | UAB | 72.39 | 27.23 | 970 |
| Qassimiut | QAS_U | UAB | 61.18 | 46.82 | 900 |
| Tasiilaq | TAS_U | UAB | 65.67 | 38.87 | 570 |
| Nuuk | NUK_U | UAB | 64.51 | 49.27 | 1120 |
| Upernaviq | UPE_U | UAB | 72.89 | 53.58 | 940 |
| Thule | THU_U | UAB | 76.42 | 68.15 | 760 |

Surface type UAB: Upper-Middle Ablation zone, ACC: accumulation area.

## 2.3 IceBridge

The Operation IceBridge project (Kurtz et al., 2013) conducts flight campaigns over the Arctic sea ice and the GIS, carrying various instruments amongst which a thermal infrared radiometer, KT19, which observes in a similar IR frequency interval as the AHVRR Channel 4 (9.6-11.5 $\mu$m). Surface temperatures are retrieved by measuring brightness temperatures and assuming a constant surface emissivity of 0.97. In total, IST retrievals from 27 IceBridge flights (version 2) (Studinger, 2020) starting at the end of March and ending on May 16th 2012 were used. Due to the high resolution footprint of the KT-19 instrument, resulting in high variability of the observed radiometric surface temperature, IceBridge observations were averaged for every kilometre to make them more comparable to the lower resolution satellite data. It should also be noted that the IceBridge observations have not been screened for potential clouds. If clouds occur between the aircraft and the surface, the radiometer will observe the temperature of the (usually colder) clouds instead of the surface.

## 3 Methods

### 3.1 Level 4 OI IST

Upstream L2 observations were aggregated on a fixed grid to Level 3 (L3) and combined using a statistical methodology similar to Høyer and Karagali (2016), resulting in L4 gap-free, merged and optimal interpolated daily fields with a 0.01° latitude and 0.02° longitude resolution. Prior to the optimal interpolation (OI) an intermediate L3 super-collated (L3S) product was generated from the collation of all the L3 fields. The OI method is similar to the one from the high latitude SST DMI processing scheme (Høyer and She, 2007; Høyer and Karagali, 2016), which operates with anomalies from a first guess field. In the current approach, a persistence-based method is applied, which uses the previous analysis field as the first guess field. The IST observations from within 48 hours of the analysis time are aggregated and interpreted as anomalies with respect to



the first guess field. The OI method, given statistical input such as a first guess error variance or co-variance functions and uncertainties of the individual observations, finds the solution with lowest errors for each grid point (Høyer et al., 2014). The search radius for the OI method is set to 75 km and the maximum number of satellite observations included in the optimal estimation is 16. A spatially and temporally varying dynamical bias correction has been applied to reference the MODIS products against the AASTI GAC data (Høyer et al., 2014). This occurs when constructing the L3S product prior to generating

the L4 IST product. The temporal window for the dynamical correction is 7 days and the bias fields are smoothed over 500 km. Due to the limited temporal availability and sampling pattern (see Section 4.1), AATSR data were not used for the generation of the L4 IST product.

The satellite products used in this study represent the clear-sky IST as the IR satellite sensors cannot observe the surface through clouds. As a result, a clear-sky bias is usually observed when comparing averaged clear-sky surface temperatures

against averaged all-sky temperatures (Koenig and Hall, 2010; Comiso, 2003). Nielsen-Englyst et al. (2019) used PROMICE observations to estimate the clear-sky bias introduced when averaging using different temporal windows. Using a 72-hour averaging window, they found a clear-sky bias of -0.96°C when PROMICE stations located in the middle/upper ablation zone and the accumulation zone were used. Here, the clear-sky bias of 0.96°C has been added to the satellite products in order to provide an estimate of the corresponding all-sky daily IST fields, which can be compared to the all-sky ISTs observed by

PROMICE and IceBridge.

### 3.2 HIRHAM5 Regional Climate Model (RCM) and Surface Mass Balance (SMB) model

The HIRHAM5 RCM simulates the climate of Greenland at a 5-km resolution (Mottram et al., 2017). The RCM contains a simplified SMB model and a 5-layer snow and ice sub-surface scheme. The surface energy and water budget from the RCM, i.e. the radiative and turbulent heat fluxes and precipitation, evaporation and sublimation outputs are then used to force an

offline SMB model with more and deeper layers and a more sophisticated treatment of snowpack processes to calculate the ice sheet SMB. The full model set up is described in Mottram et al. (2017); Langen et al. (2015, 2017).

The Lagrangian set-up with 32 unequally spaced layers down to 100 m of water equivalent depth in the snow and glacier subsurface scheme is used as described in Hansen et al. (2021). Previous studies have shown that assimilating an albedo product over bare glacier ice based on MODIS data can improve ice sheet surface melt estimates (Langen et al., 2017), however

satellite-derived albedo data are not available as far back in the past and suffer from other biases. Thus, the focus in this study is exclusively on IST assimilation without albedo data assimilation but use of an internally calculated albedo scheme. The SMB model calculates the accumulation of snow and ablation based on the surface energy balance by calculating a theoretical skin temperature. Since the skin temperature of ice cannot go above 273.15 °K, i.e. the melting point, this is converted to energy available for melting surface ice. The skin temperature also influences deeper snow pack temperatures via diffusion.

Meltwater is assumed to run off straight away if bare glacier ice is exposed, but if there is snow on glacier ice, percolation into the deep layers and the associated latent heat released by refreezing in deeper layers is accounted for until the heat capacity and porosity of the layers is filled and no more percolation is possible. The sum of precipitation minus evaporation, sublimation and runoff of meltwater gives the daily SMB over the ice sheet.

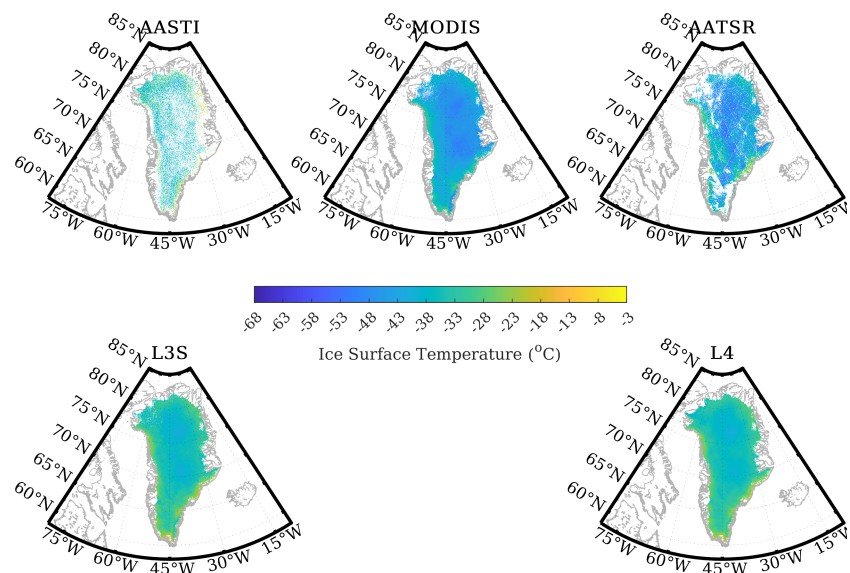

**Figure 2.** Examples of aggregated IST observations from Jan 9th, 2012 over the Greenland Ice Sheet. Top row: Level 3 AASTI (left), Level 3 MODIS on Aqua & Terra (middle), Level 3 AATSR (right). Bottom row: Level 3S (left), Level 4 Optimal Interpolated IST (right).

For the control simulation, surface energy balance outputs from the RCM were used to calculate the IST and melt potential as
normal. This control simulation was initially evaluated against the L4 IST data (not shown). For the simulation with assimilation of the L4 IST, HIRHAM5 RCM forcing was initially used to calculate IST, and if this was below -2°C at a grid point for a given time-step, the L4 IST product was assimilated. Therefore, at any given time-step, the GIS IST output by the SMB model is a combination of modelled and observed L4 IST. The threshold was chosen to filter out biases at higher temperatures observed within the L4 IST product compared with PROMICE weather station data.

**4   Results**

**4.1   Inter-comparison of IST products**

Examples of the different L3 satellite products generated from the L2 data-sets described in Table 1 are shown in Figure 2. The L3 products are aggregated for January 9, 2012 into the L3S product (bottom left) while after optimal interpolation, the L4 gap free product (bottom right) is produced for the same date. The coarser spatial resolution of AASTI (top left) compared
to MODIS (top middle) is visible, resulting in AASTI grid points with missing information while MODIS daily aggregated L3 data offer superior coverage over the GIS. The sampling of AATSR (top right) with its narrow swath and lower temporal resolution results in characteristic artefacts resembling the ENVISAT platform orbit. Such artefacts do not appear nor for the MODIS neither the AASTI products.



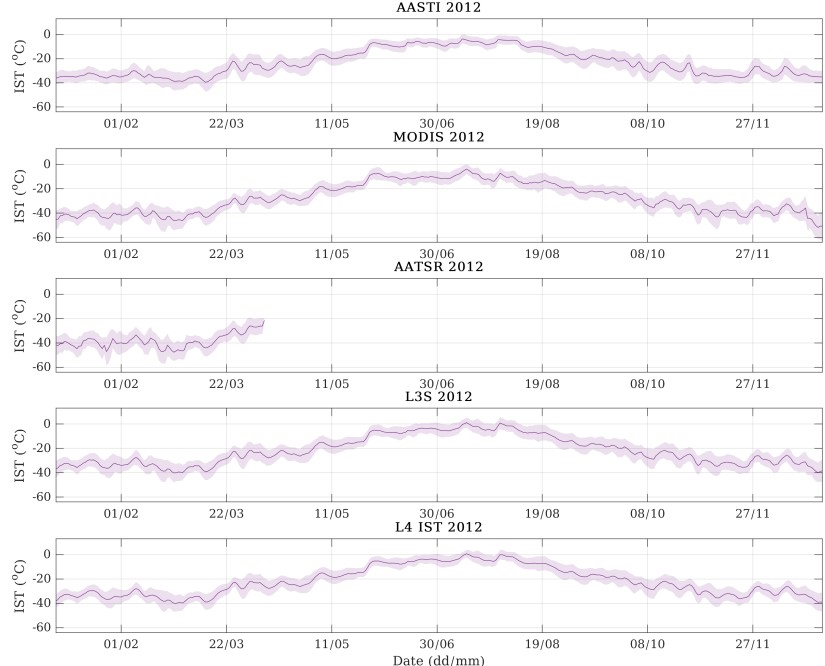

**Figure 3.** Mean daily IST (solid line) and its standard deviation (shaded area) over the Greenland Ice Sheet from the L3 AASTI, L3 MODIS, L3 AATSR (when available) and the derived L3S and L4 IST product.

Figure 3 shows time-series of mean daily ISTs and their standard deviation (shaded area) for 2012 from aggregated AASTI L3, MODIS L3, AATSR L3, L3S and L4 IST. MODIS and AATSR (when available) show colder ISTs in particular during winter and late autumn compared to the other products, with minimum MODIS ISTs of about -50°C and AASTI and the L3S and L4 IST products reaching their lowest ISTs of -35°C to -40°C. All products, including the L4 IST, well represent the annual cycle with the warming that started in early March and peaked in July, followed by cooling and winter minimum at the end of December.

The mean monthly IST and its standard deviation for all L3 products and the L4 IST are shown in Figure 4. Although AATSR was only available until the beginning of April, a monthly value was calculated nonetheless. From January to March, mean monthly ISTs from MODIS (magenta) and AATSR (green) were similar and significantly lower than AASTI (blue), the L3S (cyan) and the L4 IST product (red) accompanied by a higher standard deviation. These differences decreased from April to June, yet MODIS consistently showed lower values and higher variability compared to AASTI and the derived L3S and L4 products, which consistently agreed throughout the year. All products showed a peak mean monthly value in July, while June was warmer than August. From January to March, mean monthly temperatures were comparable to November-December means and standard deviations were of the same order.

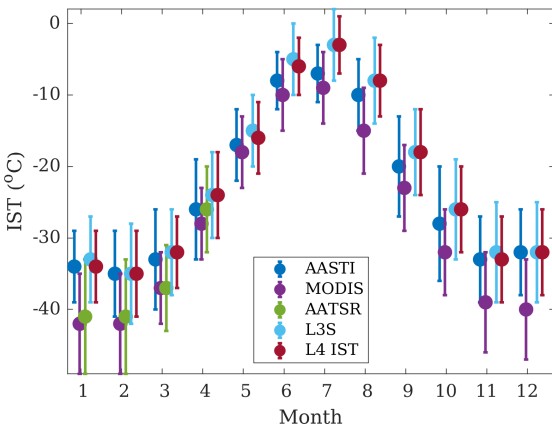

**Figure 4.** Mean monthly IST (dots) and its standard deviation (bars) over the Greenland Ice Sheet from the L3 AASTI, L3 MODIS, L3 AATSR and the derived L3S and L4 IST product, when available.

**Table 3.** Mean annual and seasonal IST with standard deviation in °C, from the different L3 and the L4 IST products for the Greenland Ice Sheet for 2012.

|  | Winter (DJF) | Spring (MAM) | Summer (JJA) | Autumn (SON) | Annual |
|---|---|---|---|---|---|
| AASTI | -33.9±5.9 | -25.1±6.3 | -8.2±4.3 | -26.8±6.9 | -23.4±5.9 |
| MODIS | -41.2±6.9 | -27.8±5.2 | -11.5±5.2 | -31.5±6.4 | -27.9±5.9 |
| L3S | -33.4±6.4 | -24.0±5.8 | -5.4±5.2 | -25.1±6.6 | -21.9±6.0 |
| L4 IST | -33.6±5.7 | -24.1±5.4 | -5.5±4.5 | -25.3±6.1 | -22.1±5.4 |

For all products, December 2012 was used to estimate the mean winter IST value. AATSR was excluded as data were only available unti April.

Such differences and variabilities are also reflected in the mean seasonal and annual estimates, shown in Table 3 for the different L3 products (excluding AATSR) and the L4 IST product. AASTI, L3S and L4 IST showed higher, and similar, seasonal and annual means compared to MODIS. Discrepancies between the estimates ranged from 0.5°C in winter to ∼1°C in summer, between AASTI, the L3S and L4 IST products. MODIS was 3°C-8°C colder, with discrepancies being smallest in spring. Mean annual IST over the GIS ranged from −22.1±5.4°C for the L4 IST to −27.9±5.9°C from MODIS.

The primary reason for the lower LST_cci v1.0 MODIS and AATSR IST values, used in the present study, is the type of cloud masking applied in the first version of the data. No post-filtering or implementation of the cloud masking techniques (later developed within the LST_cci for both instruments) were applied in the v1.0 of the data presented here but only the standard operational cloud mask. For the case of AATSR, in addition to the cold bias, there also was the sampling issue (see Figure 2) and the limited availability of data for the reference year 2012 (contact with ENVISAT was lost in April). Therefore, AATSR was not used to generate the final L4 IST product. With respect to the MODIS product, the pixel-to-pixel variability





is smaller than for AASTI, mostly associated with the better coverage (see Figure 2) and it was thus decided to use it for the
generation of the L4 IST product.

## 4.2    Validation of IST products

Figure 5 shows aggregated daily mean biases and standard deviations for AASTI, MODIS, L3S and the L4 IST product
against the PROMICE stations (see Figure 1 and Table 2). Note that, for each day, all available PROMICE stations were used
and that number differs from day to day, as not all stations may have availability for all days. The overall bias of AASTI
data was −2.16°C±3.89°C, significantly smaller than −11.59°C±7.48°C reported for MODIS. The L3S product bias was
−4.30°C±5.80°C while the L4 IST product was also found cold with a mean bias of −4.04°C±5.15°C, i.e. 0.26°C lower bias
and 0.65°C lower standard deviation compared to the L3S product. During summer, mean bias and standard deviation for
AASTI was very stable and close to zero which was not the case for the MODIS data. The L3S and L4 IST products also show
higher variability in the daily bias and standard deviation during summer, compared to AASTI, yet they appear more stable
than MODIS, indicating the benefits of using the the OI methodology to generate daily, gap-free IST fields. Furthermore, the
lower bias and standard deviation of the L4 IST product against PROMICE stations, along with its better coverage demonstrate
its potential advantage over single sensor or daily aggregated fields.

When examining the performance of the L3 and L4 products individually for each PROMICE station (Figure 6), AASTI
consistently had lowest bias (from −3.24°C to −1.34°C) and standard deviation values (3.29°C-5.83°C) for all stations, in-
dependent of altitude, type of ice sheet zone and geographical location. MODIS, beyond higher bias and standard deviation
values also showed higher variability, with biases ranging from −17.43°C to −5.97°C and standard deviations between 6.41°C
and 9.39°C. Such behaviour provides additional evidence that the cold-bias issue stems from the early version of MODIS data
used in this study and the cloud masking approach implemented.

Other studies have shown similar cold biases in surface temperatures derived from MODIS (Wan et al., 2002; Hall et al.,
2012, 2013). However, as shown in Section 4.1, the important contribution of the aggregated MODIS observations in achieving
a proper coverage over the GIS justifies the dynamical bias correction of the MODIS product against the AASTI data described
in Section 3.1 and its further use for the generation of the L4 IST product.

This higher variability of the MODIS product influenced the performance of both the L3S and the L4 IST product, where
biases ranged from −8.61°C to −0.62°C and standard deviations from 3.68°C to 7.42°C. The overall performance using
the PROMICE stations was −2.22°C±4.32°C for AASTI, −11.9±8.17°C for MODIS, −4.58°C±6.47°C for the L3S and
−4.31°C±5.78°C for the L4 IST product. The stable performance of AASTI and the L4 IST indicated that no single station
and its characteristics (location, altitude) influenced the validation statistics.

Using the 1-km averaged surface temperature observations from 27 IceBridge flight campaigns, starting in March 30 and
ending in May 16, mean bias (dots) and standard deviation (vertical bars) were estimated and are shown in Figure 7 for AASTI,
MODIS, L3S and the L4 IST. For most flights, AASTI observations were in agreement with the IceBridge observations, with
overall zero bias (−0.01°C±4.03°C) and a very stable behaviour as no major outliers occurred; biases ranged from −2.10°C to
2.10°C. MODIS was cold compared to the flight measurements, manifested as a negative bias (−5.19°C±4.8°C), and with a





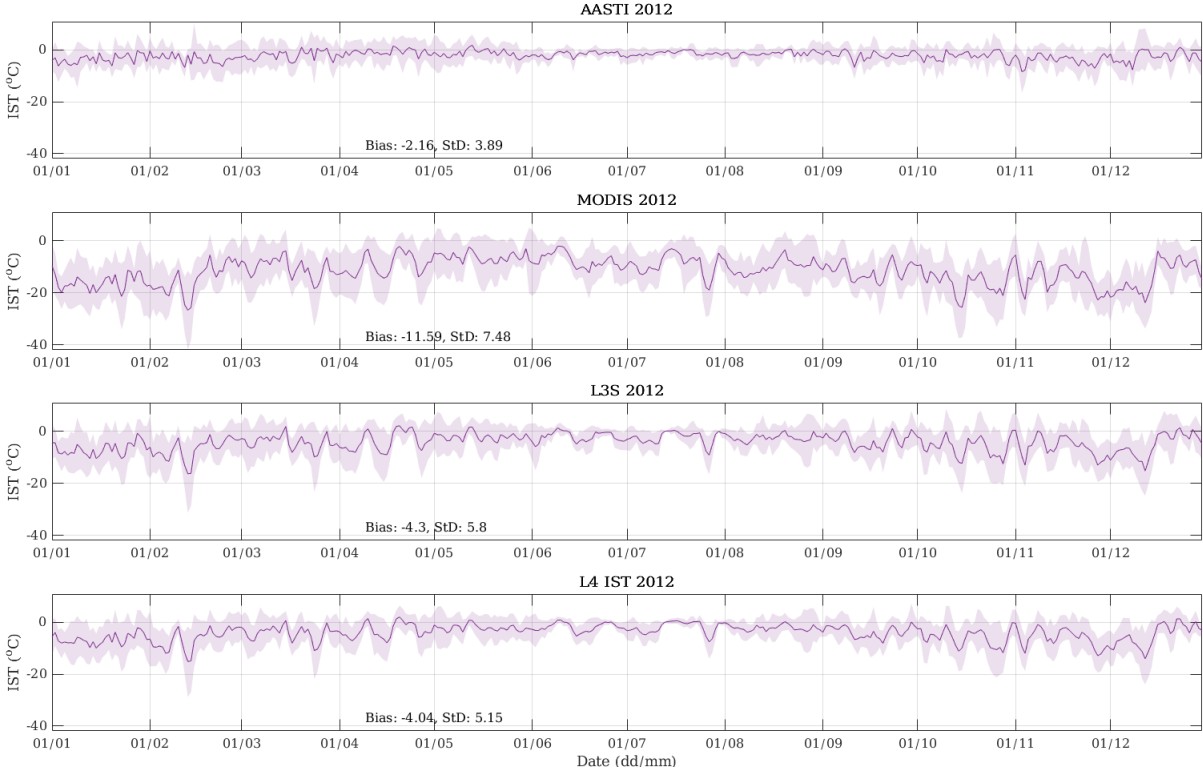

**Figure 5.** Time-series of mean daily bias estimates (line) and their standard deviation (shaded area) for AASTI, MODIS, L3S and L4 IST against aggregated in situ observations from the PROMICE stations.

pronounced variability during the period evident from the oscillating bias (from −14.15°C to 2.20°C) and standard deviation values (from 2°C to 7.2°C).

The L3S and L4 IST products (bottom panels of Figure 7) showed significantly lower bias and standard deviation compared to MODIS although with a similar, yet reduced variability in the statistics depending on the campaign; biases ranged from −6.66°C to 3.98°C and standard deviations from 1.7°C to 6.6°C. Overall, the L3S product had a bias of −0.63°C±4.27°C and the L4 IST a bias of −0.61°C±3.59°C, i.e. the lowest standard deviation of all data sets considered, ∼0.44°C and ∼0.6°C lower than the AASTI and L3S product, respectively. As was the case for the PROMICE comparisons, the L4 IST product combined

stability and robustness in its validation performance and along with its improved spatial coverage can be considered more relevant for applications, e.g. related to SMB modelling of the ice sheet.

An example of one IceBridge flight campaign is shown in Figure 8 for April 19th 2012. The top panel shows the L4 IST product along with the flight path, marked as a line indicating distance, starting from the darkest colour at 0 km and progressing towards the lightest shades at the end of the flight (>3500 km). Flight measurements averaged every one kilometre as a function

of distance covered during the flight (magenta) and their standard deviation (shaded area) are shown in the bottom panel along



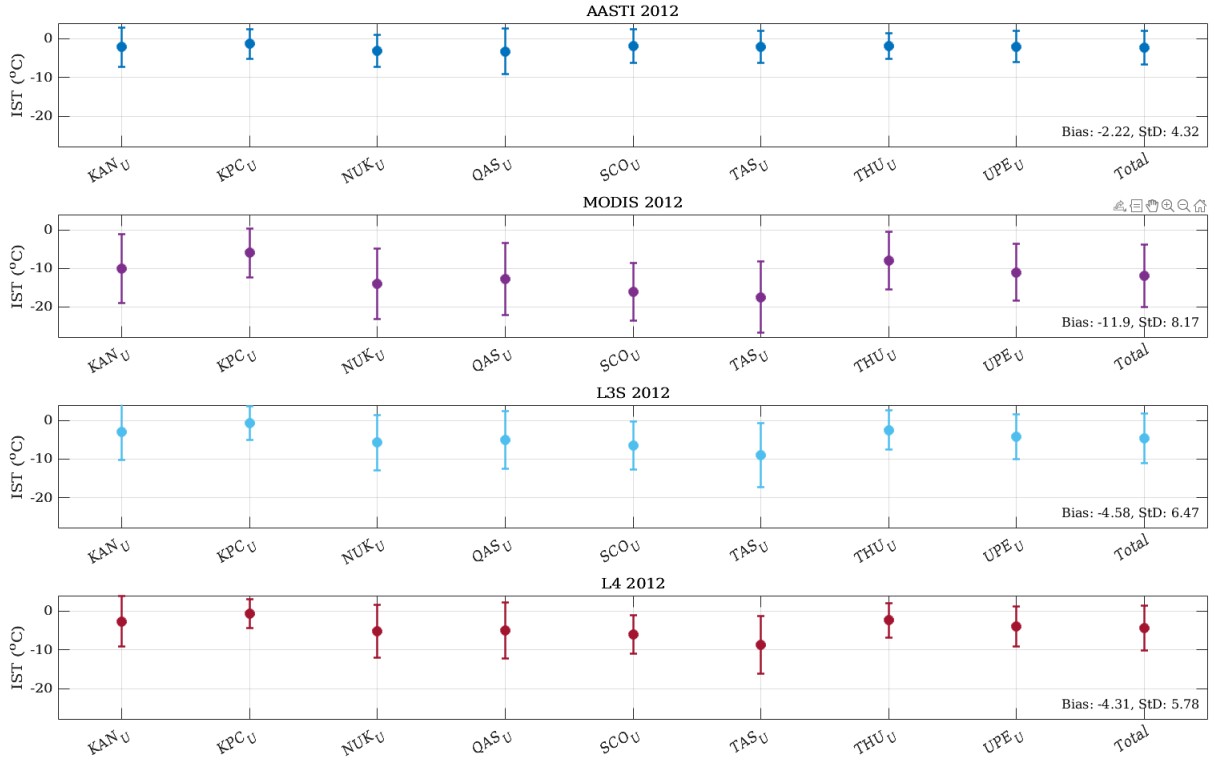

**Figure 6.** Mean bias (dots) and standard deviation of errors (bars) for AASTI, MODIS, L3S and L4 IST against in situ observations for each of the PROMICE stations.

with values from the L4 IST product, extracted for the grid points corresponding to the flight path (blue line). The mean bias for that campaign was 0.40°C±4.30°C. This campaign covered various zones of the GIS, and the variability of the IST was intense as revealed by the 1-km averaged measurements. Beyond the warm bias during the first 800km of the flight, the L4 IST captured the variability of the ISTs over the GIS remarkably well. As the IceBridge radiometer measures the radiometric

surface temperature from an aircraft at an approximate height of 450 meters above ground level without any cloud screening, the presence of clouds may explain the discrepancies for the first 800 km of the flight where the IceBridge data are significantly colder than the L4 IST.

In order to assess the impact of the high resolution footprint of the IceBridge measurements on the validation statistics, i.e. 1-km averages over 5-km averages of the L3 and L4 IST products, the standard deviation of averaging raw measurements over

different spatio-temporal windows minus the raw measurements were computed (not shown). This is an assessment of biases introduced from comparing flight data of very high resolution, i.e. resolving small-scale variability, against space-borne sensors which although are referred to between 1 km and 5 km grids, are known to resolve scales lower than their reference grid.

Using only IceBridge campaign data, averaging for different spatial windows, i.e. 1 km, 5 km and 25 km, and subtracting raw measurements, the standard deviation values for each campaign were estimated (not shown). The largest component standard




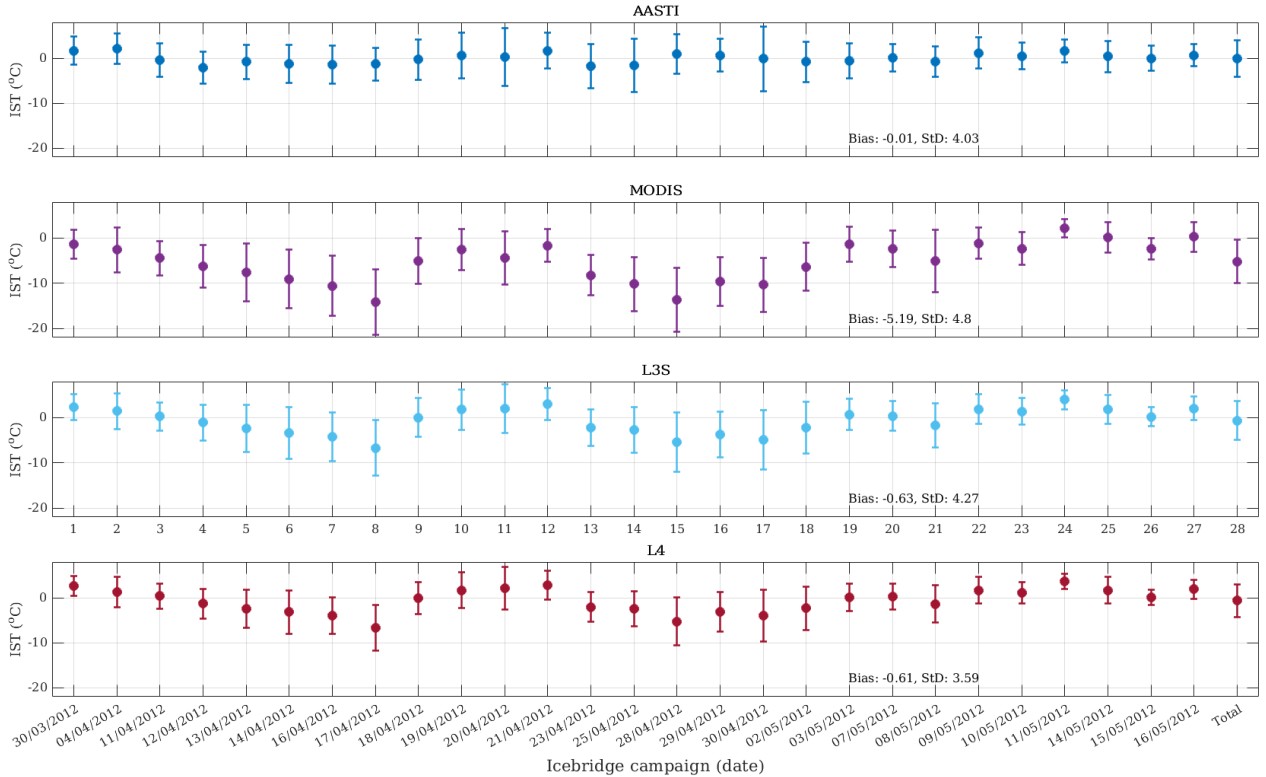

**Figure 7.** Mean bias (dots) and standard deviation of errors (bars) for AASTI, MODIS, L3 IST and L4 IST against individual IceBridge flight observations from March to May 2012. The total bias and standard deviation values are reported.

deviation was introduced when processing the raw data to 1-km averages, in the order of 2°C. Depending on the campaign, an additional 0.1°C to 0.6°C of the standard deviation was attributed to the averaging from 1 km to 5 km. When averaging from 1km to 25 km, differences in the standard deviation reached 0.9°C while the mean difference in the standard deviation over all campaigns was 0.22°C from 1 km to 5 km averaging and 0.4°C from 1 km to 25 km. Thus, between 0.2°C to 0.4°C of the standard deviation in all comparisons against IceBridge campaigns (see Figure 7) can be attributed to the different spatial

scales represented in the IceBridge data compared to satellite observations.

The validation of the AASTI, MODIS, L3S and L4 IST showed that for both PROMICE and IceBridge, AASTI had an overall better performance, with lower biases and standard deviations. MODIS data from the LST_cci project, used in this study, had a significant cold bias, associated to the less advanced cloud mask algorithm applied to the v1.0 data, which influenced the performance of the derived L4 IST product. Nonetheless, the better spatial coverage and higher resolution of MODIS rendered

it crucial for the generation of the L4 IST product and thus its inclusion was justified. A new, improved version of the MODIS L2P dataset, to be released by the LST_cci, is expected to result in better performance of the L4 IST OI product.

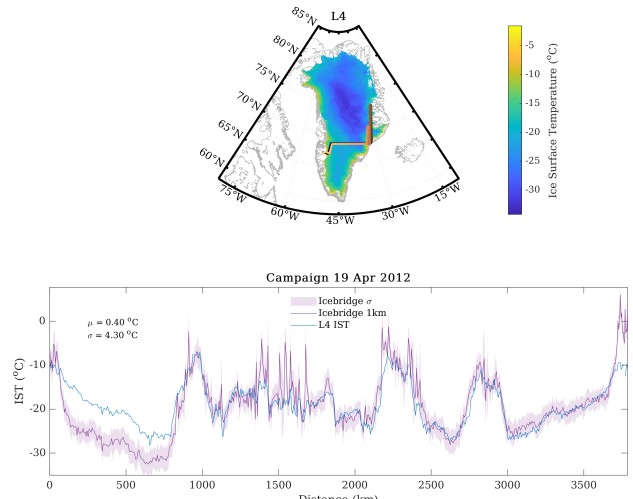

**Figure 8.** Top: L4 IST field for April 19th 2012 with dots indicating the IceBridge flight path. Darker colours indicate the start of the flight, lighter colours the end. Bottom: 1-km averaged IST from Icebridge on April 19th 2012 (magenta) and standard deviation (shaded area) along with the corresponding L4 IST values (blue).

## 4.3 Analysis of L4 IST

Monthly averages from the L4 IST product for 2012, shown in Figure 9, indicate the extent of warming during the summer months where temperatures on the GIS ranged from −16° to zero, especially during July and for most of the ice sheet. Similar ranges were reported in Hall et al. (2008) for the period 2000-2006 from MODIS LST products as well as in Hall et al. (2012), using daily MODIS IST between 2000 and 2010.

The number of aggregated observations from AASTI and MODIS used to generate the L4 IST product, shown in Figure 10, revealed a seasonal pattern where most observations were available between May and August while fewest observations were available from November to February. The northern part of the GIS was consistently observed fewer times compared to the central part. This spatial and temporal variability in the availability of observations from MODIS and AASTI is related to the availability of observations in the infra-red which is limited by cloud cover.

The annual mean IST over the GIS for 2012 is shown in Figure 11 (left panel), using the L4 IST product. Values ranged from −30°C for the central part of the ice sheet and dropped down to −8°C for the terminal zones, particularly for latitudes south of 70°N. Hall et al. (2008) reported similar values using MODIS data between 2000 and 2006. Furthermore, the mean IST during the melt period May-August 2012 (Figure 11, middle) computed from the L4 IST product showed values ranging from −15°C and up to 0°C, in agreement with Hall et al. (2008).

Melt days were defined as days for which the IST was −1°C or higher, following Hall et al. (2013). They were estimated for the period May 1st to August 31st, at each grid point over the GIS. Figure 11 (right panel) shows the number of melt days from the L4 IST product, where white areas experienced zero melt days and coloured areas indicate at least one melt day or more.



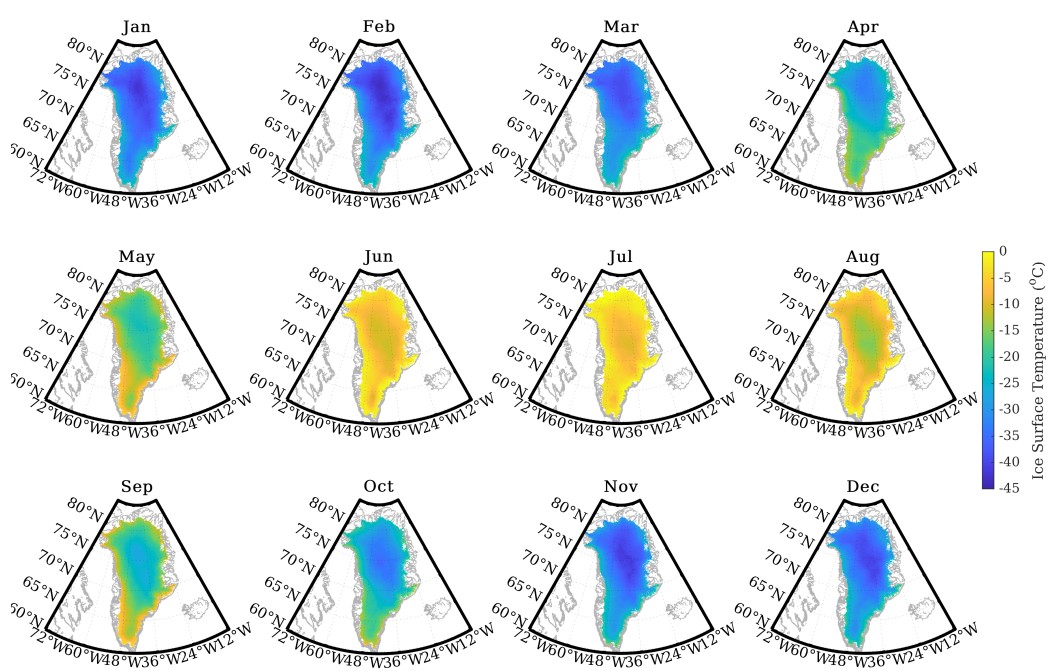

**Figure 9.** Mean monthly ice surface temperature over the Greenland Ice Sheet for 2012, from the L4 IST product.

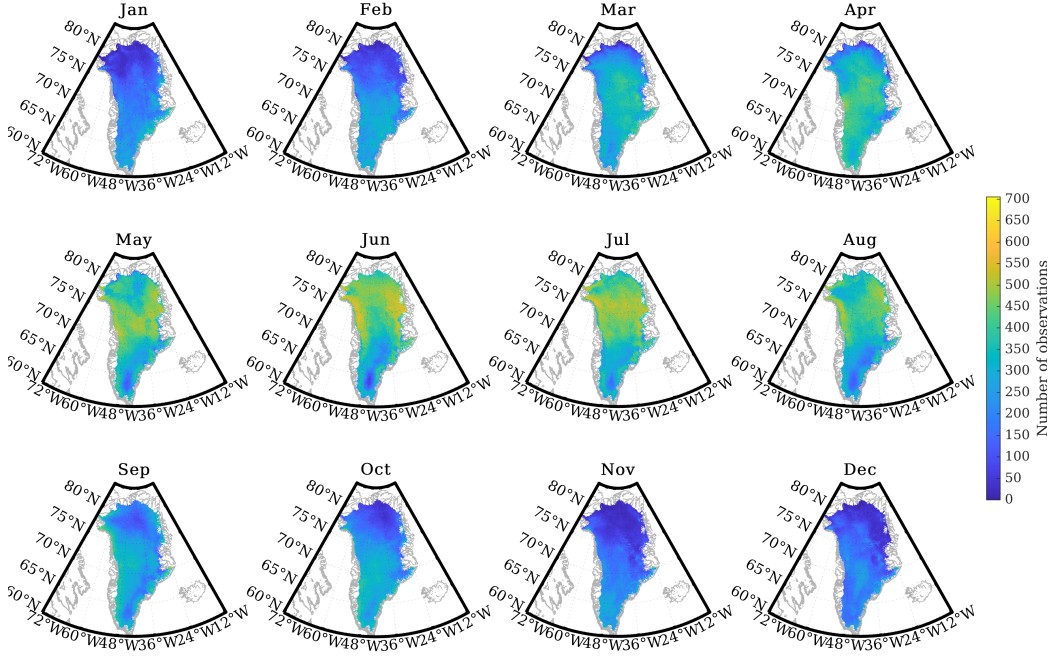

**Figure 10.** Number of aggregated observations from MODIS and AASTI used for the daily L3S and L4 IST products.





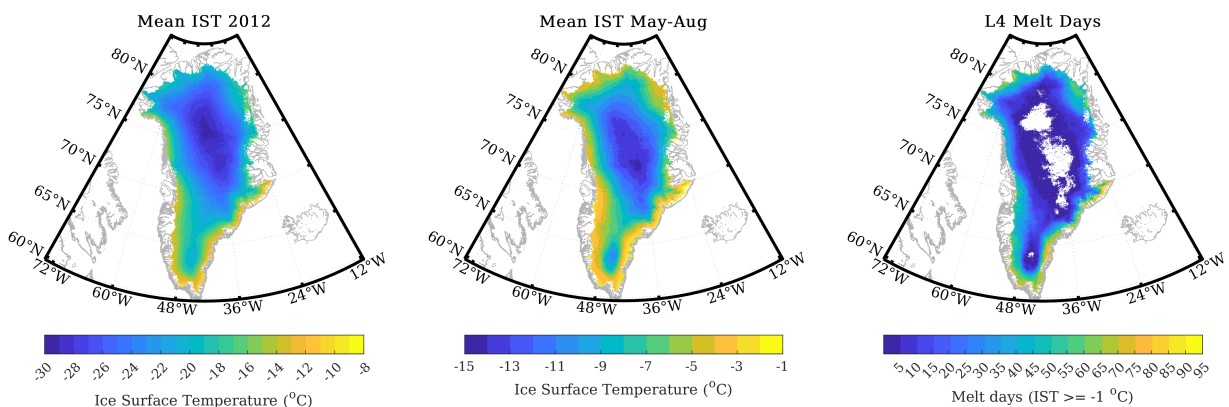

**Figure 11.** Mean IST for 2012 (left), for the melt season May-August 2012 (middle) and number of melt days during the melt season (right, minimum 1 day, white areas on the GIS indicate zero days), from the L4 IST product.

Melting was observed over large parts of the GIS for more than one day while significant parts of the middle and lower zones experienced more than 30 days of melt conditions.

## 4.4  IST Assimilation experiment

Mean monthly IST values for May were estimated from the control (left) and updated simulations including assimilation of the L4 IST product (middle), shown in Figure 12, along with the estimated anomaly (right). The month of May was selected

since it is the month when the onset of melting commonly occurs across much of western and southern Greenland; this is a challenging period for SMB models to simulate and the use of IST observation can potentially have a positive impact.

The updated simulation using assimilation of L4 IST, was generally warmer over a large part of the GIS, especially the east and north-east regions. The difference between the two mean May estimates, computed by subtracting mean May estimates of the control simulation from the one using assimilation of the L4 IST (right panel in Figure 12) highlighted the areas for which

the control simulation was consistently colder by 2°C and more, even up to 5°C, extending to the north, south and east parts of the GIS. To the contrary, the control simulation showed warmer temperatures on the west and central part of the GIS yet differences in this case did not exceed 1°C and only for small areas were they up to 3°C.

Comparing the simulations against the PROMICE stations for May 2012 (Figure 13) showed that mean daily temperatures from both the control (top) and updated simulation, using assimilation of the L4 IST product (bottom), were colder compared to

PROMICE station measurements. The bias was lower for the updated simulation (−1.14°C) compared to the control simulation (−2.16°C) yet the standard deviation was higher, 3.55°C for the updated simulation compared to 2.9°C for the control. Only at the TAS_U station did both simulations indicate warmer surface temperatures, but this needs to be assessed cautiously given the very few observations available in May 2012 at this station (not shown).

Comparison with the IceBridge flight campaign measurements to assess the control (top) and updated simulation (bottom)

with assimilation of the L4 IST product (Figure 14) showed a marked improvement with the assimilation of L4 IST data, with





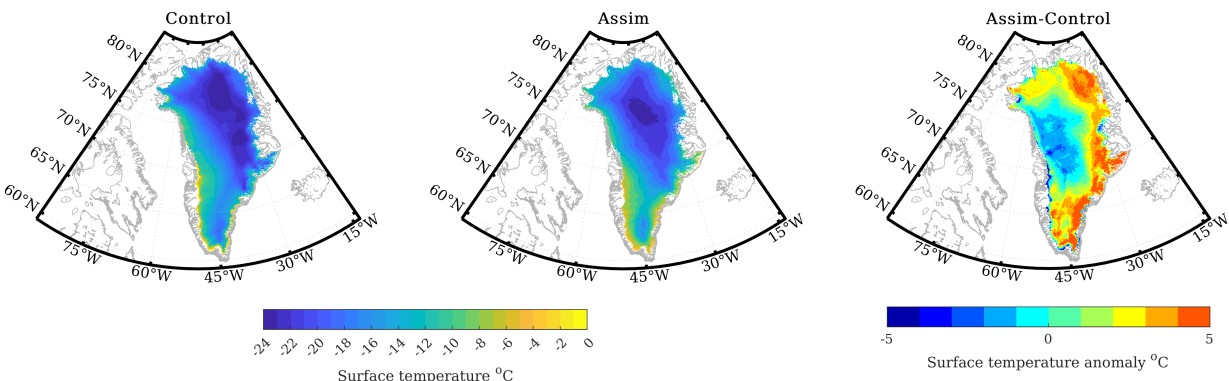

**Figure 12.** Mean monthly surface temperature for May 2012 from the control (left) and updated simulation experiment (middle) along with the anomaly (right).

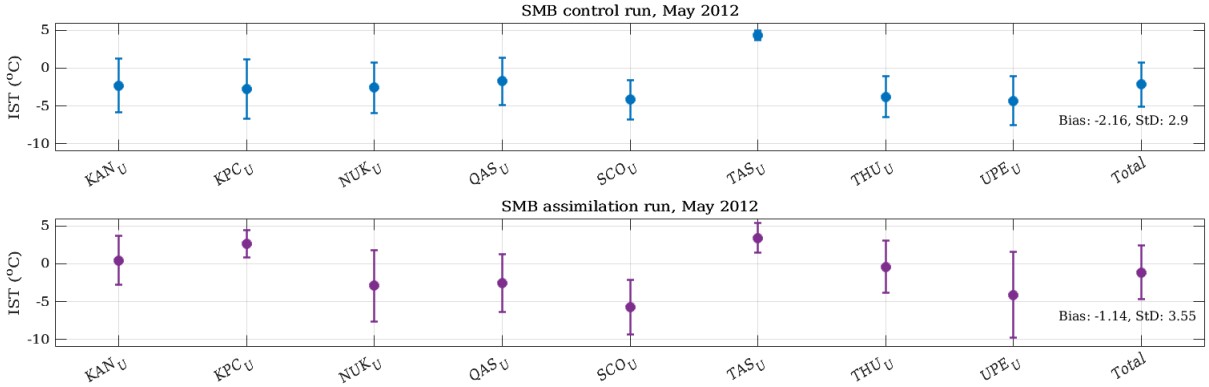

**Figure 13.** Mean bias (dots) and standard deviation (bars) for the control simulation (top) and the one using assimilation of L4 IST (bottom) against the individual PROMICE stations for May 2012.

a reduction in both the bias and the standard deviation compared to the control run. The reduced standard deviation for the IceBridge data comparison, as compared to PROMICE observations, was consistent with what was reported in Section 4.2.

An example of one IceBridge flight and associated SMB simulations is shown in Figure 15, similar to what was shown for the L4 IST in Figure 8. The top panel shows the SMB updated simulation along with the flight path, marked as a line 310 indicating distance, starting from the darkest colour at 0 km and progressing towards the lightest shades at the end of the flight (>3500 km). Flight measurements averaged every one kilometre as a function of distance covered during the flight (magenta) and their standard deviation (shaded area) are shown in the bottom panel along with values from the SMB model control (green) and updated simulation using assimilation of the L4 IST product (cyan), extracted for the grid points corresponding to



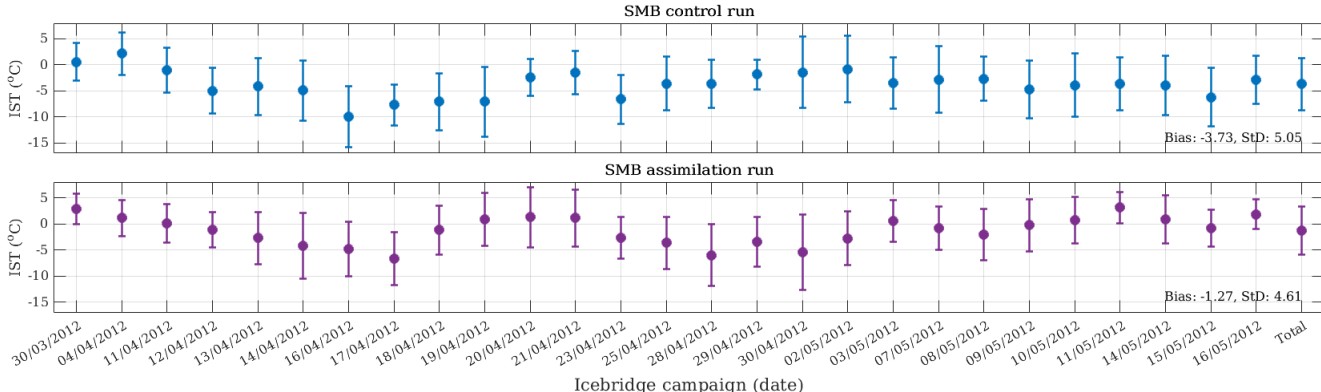

**Figure 14.** Mean bias (dots) and standard deviation (bars) for the control simulation (top) and the one using assimilation of L4 IST (bottom) against the IceBridge flight campaigns.

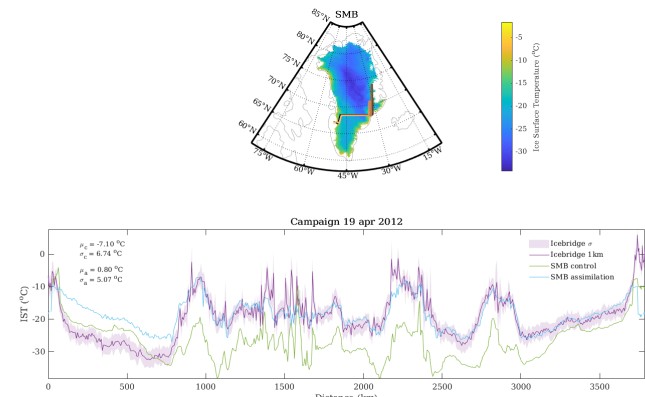

**Figure 15.** SMB simulation for April 19th 2012 with dots indicating the IceBridge flight path. Darker colours indicate the start of the flight, lighter colours the end. Bottom: 1-km averaged IST from IceBridge on April 19th 2012 (magenta) and standard deviation (shaded area) along with the corresponding values from the control (green) and updated simulations (cyan).

the flight path. For this specific flight, the mean bias and standard deviation with respect to the IceBridge measurements was
$-7.10°\pm6.74°$C for the control simulation and $-0.80°\pm5.07°$C, for the updated one.

The L4 IST product was shown to improve the SMB model performance when simulated surface temperatures where compared with PROMICE stations and IceBridge flight campaign measurements, for the test case of May 2012, selected as the typical onset of the melting season. Experiments for other months of 2012 (not shown) indicated a neutral impact of the L4 IST product, providing more confidence in the existing SMB internal parameterisations, for the least challenging periods.





## 5    Discussion

Assessing mean annual IST values over the GIS for 2021, Hall et al. (2013) reported mean annual IST of −23.62±6.24°C from MODIS, which is closer to the L4 IST values, compared to what is derived from MODIS in this study. Hall et al. (2013) manually removed daily MODIS IST fields when the cloud mask erroneously identified the ice surface as cloud free - particularly occurring during the summer. Their 2012 summer season mean IST was −6.38±3.98°C, which is significantly higher than the −11.5±5.2°C found for MODIS in the present study, and closer to the −5.5±4.5°C for the L4 IST product.

Validation of the upstream input datasets and the derived L4 IST indicated larger differences between the satellite products and PROMICE measurements during winter, which can likely be associated to the higher diurnal variability in IST during winter (Nielsen-Englyst et al., 2019) and the fact that cloud masking algorithms can suffer from reduced skill to identify cloudy from clear-sky pixels over ice-covered surfaces during the polar winter season (Dybkjær et al., 2012).

The validation of satellite IST products against in situ observations suffers from the lack of available fiducial reference observations for the IST over the GIS. As discussed in Høyer et al. (2017), the use of point-wise in situ observations can introduce a sampling uncertainty ranging from 0.4°C to 5°C, depending on the type of in situ observations, when compared to satellite observations with a 1km footprint. Such contributions are not related to the performance of the satellite products but arise from the spatial and temporal sampling uncertainty and the type of in situ observations. For the types of in situ observations used in this validation, it is expected that the PROMICE broadband IST observations will have higher uncertainties compared to the IceBridge data. This is a consequence of the PROMICE broad band radiometer observations versus the narrow band IceBridge observations, where the snow and ice surface emissivity effects, due to surface properties or incidence angles, vary much more for the broadband observations compared to the narrow band radiometer. Also, the Spectral Response Functions (SRFs) of the IceBridge KT19 instrument are very similar to the actual IR satellite SRFs. Therefore, the results from the inter-comparison should not be viewed as an estimate of the uncertainty of the satellite products.

Beyond the differences in the broad band PROMICE versus narrow band IceBridge measurements, another reason for the reduced standard deviation can be associated to the fact that the IceBridge flights also cover the interior of the ice sheet, where temperatures are lower and there is little melt. The PROMICE stations lie mostly within the ablation zone where there are large diurnal and seasonal changes in IST that are challenging for both model and satellite observations to characterise.

Analyses of the annual spatio-temporal variability of IST over GIS revealed extended warming during the summer months of 2012, already reported as an extreme melt year (Box et al., 2012; Nghiem et al., 2012). Both Hall et al. (2008), for the period 2000-2006, using MODIS LST products as well as Hall et al. (2012), using daily MODIS IST between 2000 and 2010, reported similar ranges of IST values over the GIS as found in this study. Furthermore, the 2012 annual mean IST over the GIS reported in this study (see left panel of Figure 11), is similar to what was derived in Hall et al. (2008) using MODIS observations for 2000-2006. For the melt period defined between May and August 2012, mean IST from the L4 IST product (middle panel of Figure 11) was found in agreement with Hall et al. (2008) (see their Figure 2), although for that study not ice-covered land surface temperatures where also considered thus above zero values were included.



Hall et al. (2013) reported more than two melt days for most of the GIS during the melt season of 2012, based on MODIS data, which also indicated the warmest summer in the MODIS record with mean IST of −6.38±3.98°C, in good agreement

with the mean summer IST from the L4 IST product reported in Table 3. In Nghiem et al. (2012), extended melt over the GIS was identified from a combination of space-borne sensors, including MODIS. Hanna et al. (2014) associated the reported extreme melt event of 2012 with unusually high geopotential heights and atmospheric pressure anomalies over the ice sheet.

As the year 2012 was significant in terms of melting over the GIS, the month of May specifically poses a challenge, as it signifies the onset of the melt season, i.e. a challenging period for SMB models to simulate correctly as biases in winter

accumulation can lead to significant discrepancies between observed and simulated melt onset. Hermann et al. (2018) showed that overestimates of winter snow close to the ice sheet margin delayed the onset of simulated melt compared to reality in southern Greenland, with the reverse bias being found higher up where snow fall rates were underestimated. The use of IST data to mitigate against this bias therefore has potential to improve annual melt and SMB estimates.

Both the control and updated simulation using the L4 IST product were forced by the HIRHAM5 RCM, which provides

the required 6-hourly forcing fields and allows for calculation of surface temperature at a higher temporal resolution; this also captures the diurnal cycle, likely an important process to resolve in the ablation zone, especially in spring. The coarser temporal resolution of surface temperature assimilated into the model in the updated simulation using the L4 IST, compared to the control run, can be considered a challenge for assimilating the daily L4 IST products. This is a topic that needs careful consideration when aiming to generate optimally interpolated IST fields from space-borne sensors for the purpose of providing

input to models that are typically in need of temporally resolved parameters with a frequency higher than daily. Nonetheless, the high spatial coverage offered by the L4 IST product is an attractive and important attribute when aiming to capture and represent variability over large areas of the GIS, as demonstrated in Section 4.3, that can not be resolved through in situ measuring stations and is typically under-represented in large (global) and medium (meso) scale model simulations.

## 6  Conclusions

The ESA LST_cci L2 MODIS Aqua/Terra data were used along with AASTI AVHRR GAC data to generate daily, gap-free, optimally interpolated L4 IST composites for the Greenland Ice Sheet (GIS) for the year 2012, chosen due to the extreme melt conditions. The upstream satellite data and the newly derived L4 product were validated using PROMICE and IceBridge observations. Furthermore, the L4 IST product was used to assess mean monthly, annual and seasonal IST over the GIS and provided the basis for estimating melt during 2012.

Comparisons against the PROMICE stations and airborne IST observations from the IceBridge flights, suggest that the LST_cci MODIS data are cold-biased by several degrees. This was attributed to the cloud masking algorithm used for the generation of the v1.0 MODIS data within the LST_cci, as no post-filtering or techniques (later developed by the LST_cci) were implemented. The equivalent AASTI AVHRR data did not exhibit such a cold bias and this demonstrates the importance of the multi-sensor inter-comparisons. After implementing a bias-correction to the MODIS data, agreement between the derived

L4 IST data and PROMICE stations and airborne IST measurements improved, but a residual cold bias was still evident in





the L4 IST product. This suggests that the large number of MODIS observations included in the generation of the L4 IST, compared to AASTI, might challenge the bias adjustment scheme and that an improvement could be made regarding this in a future development. In general, we suggest that dedicated improvements on the IST retrievals and cloud masking algorithms could reduce both the bias and the large regional errors presented in this study.

The larger biases and standard deviations identified for all satellite products against PROMICE stations compared to the ones for the IceBridge campaigns, were associated with the higher uncertainties of the broad band radiometers used on the PROMICE stations. The IceBridge campaigns used a narrow band radiometer, whose spectral response functions are very similar to the ones from thermal infra-red satellite instruments. We also suspect that the PROMICE stations location in the ablation zone of the ice sheet means larger diurnal variability in terms of surface energy budget that is not necessarily captured 395 by a daily data product.

By combining upstream IST satellite products, the gap-free, daily L4 IST product exhibited a stable, high quality performance when compared to the PROMICE stations and IceBridge flight measurements. Thus, advantages from AASTI, i.e. accuracy, stability and robustness, and MODIS, i.e. spatial resolution and coverage, were inherited in the L4 IST product. This allowed for a thorough analysis of IST spatial and temporal variability over the GIS during the challenging year of 2012. 400 Findings were in agreement with other studies, which were nonetheless based on single sensor satellite products. The L4 product is thus useful for understanding larger spatial and temporal variability over the GIS, not achievable using limited, local measurements or single-sensor satellite observations.

L4 IST daily fields were also ingested into an SMB model, forced by outputs from a regional climate model, to estimate ice melt and retention. The impact of using observed IST data in the model was assessed by comparing modelled and observed 405 estimates of the surface temperature for 2012 when extreme melting occurred. A major challenge in this approach was the degradation in temporal resolution of surface temperature by forcing the assimilation of daily IST values from the gap-free L4 IST product. Nonetheless, for the melt onset period of May 2012 it was found that assimilating the daily L4 IST product produced more realistic surface temperatures, when compared to PROMICE stations and IceBridge flight campaigns. This suggests that, while a continuous forcing of the SMB model with daily L4 IST may not provide significant improvements 410 at all locations and times of the year, at least when the temporal resolution of satellite IST products is daily, allowing for assimilation of the product during challenging periods of the year, e.g. onset and during the melt season, can improve SMB estimates. Furthermore, significant value in the L4 IST dataset as a means to evaluate climate and SMB models and to conduct process studies was identified. The finding that even with the coarser temporal resolution, the data assimilation improves surface temperatures over the large interior of the ice sheet is in itself an important result. Sensitivity studies, e.g. changing the 415 timestep of assimilation and accounting for the time of IST data acquisition, are likely to further improve the use of IST data in weather and climate models.

*Data availability.* PROMICE data are available from http://www.promice.dk. Operation IceBridge data are available from https://www.nasa. gov/mission_pages/icebridge/index.html. ESA LST_cci data are available from the JASMIN facility http://gws-access.jasmin.ac.uk/public/





esacci_lst/. AASTI AVHRR GAC data are available upon request. The generated L4 IST dataset is to be distributed through the dedicated
420 ESA LST_cci repositories, pending upload. SMB model simulations are available on zenodo (DOI to be confirmed).

*Author contributions.* JLH and RM devised the initial concept for this study. DG provided the ESA LST_cci data and final comments. JLH
designed and produced the L4 IST. RM designed and executed the simulation experiments. IK performed the inter-comparison of L4/L3 &
analysis of L4 IST. MBS and PNE performed the validation of L3/L4. MBS prepared inputs for and analysed the simulations. IK led the
authoring of the manuscript with contributions from MBS, RM, PNE, GD and JLH. All authors have read and agreed to the published version
425 of the paper.

*Competing interests.* Co-author Dr Ruth Mottram is an editor for TC.

*Acknowledgements.* Funding from the ESA CCI+ Phase1 New ECVS LST (ESA/Contract No. 400123553/18/I-NB) has been used in this
study. The study was also funded by The Danish National Centre for Climate Research (NCKF) at the Danish Meteorological Institute.





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
