# Peer review of "A new L4 multi-sensor ice surface temperature product for the Greenland Ice Sheet"

_The Cryosphere, 2021_

## Referee Comment (RC2)

[referee-annotated manuscript omitted]

---

## Author Comment (AC1)

**Response to comments from Referee #1 for TC-2021-384**

Dear reviewer, Please see the response to your input below.

**1 General Comments**

- *This paper presents first a new data set of ice surface temperature (IST) merging different satellite products and assimilates afterwards this new IST data set to "improve" SMB estimations over the Greenland ice sheet. While the development of such L4 data sets is innovative and scientifically relevant as well as the assimilation into a SMB model, some improvements are needed before potential publication in TC.*

  We thank the reviewer for reading the manuscript and providing constructive comments.

**2 About the L4 data set**

- *Line 122: Why a radius of 75km? What is the sensibility of using a larger/lower radius?*

  The choice of 75 km search radius was a compromise between selecting a large search radius that ensured enough data to be included in the OI estimate and a computationally feasible search grid box. This search radius is also used in the CMEMS NRT L4 SST/IST product. The key point for this study is that the average number of observations found within the 75 km is very close to the maximum number of observations, which implies that the search radius is not severely limiting the number of observations available for the OI. This has been included in the text, Lines 360-365: "The choice of 75 km for the search radius of the Optimal Interpolation scheme was a compromise between selecting a large search radius that ensured enough

data to be included in the OI estimate and a computationally feasible search grid box. The average number of observations found within the 75 km search radius is near the maximum number available of observations, thus the selected search radius is not severely limiting the number of observations available for the OI. The threshold of 75 km is being used for the operational production of the Near-Real-Time Arctic Ocean L4 SST/IST (Copernicus Marine Service, 2021)."

- *Fig 3. Like fig4, all the time series should be shown on the same plot to better allow to compare the different data. Here, in view of the used -60°C +10°C vertical axis, it is impossible to evaluate if a product is warmer/colder than an other one. Moreover, nothing is said about the considered area. Is it the same one for each product or is it an integrated value over available area in each data set (which could be very different). A plot using at least the same area for each day and each data set should be built. Finally, the differences in the passing time should be evaluated. The differences shown in Fig 2 could be due to the passing time which is different in each data set and therefore, these figures can not be compared for me.*

We did combine time-series of mean daily values and associated standard deviation in one plot, yet it is actually harder to interpret as lines and shaded-areas overlay making it very hard to understand which is which (see figure below, which nonetheless excludes the L3S product). Therefore we refrain from modifying Fig.3 and we keep the y axis limits the same for all panels, so it can be straightforward to compare. For example, using the consistent y-axis limits one can see that for winter mean MODIS and AATSR values are below -40°C while AASTI and the L3/L4 are above. Such guidance is also provided in the text, see lines 180-183: "MODIS and AATSR (when available) show lower ISTs in particular during winter and late autumn compared to the other products, with minimum MODIS ISTs of about -50°C and AASTI and the L3S and L4 IST products reaching their lowest ISTs of -35°C to -40°C. All products, including the L4 IST, well represent the annual cycle with the warming that started in early March and peaked in July, followed by cooling and winter minimum at the end of December."

Regarding the considered area used to derive mean daily temperatures in Figure 3, please see description in Lines 174-179:

[Figure]

Figure 1: All mean daily time-series combined in one figure.

"To estimate the mean daily IST for each dataset, a mask defining the Greenland Ice Sheet was applied and all valid and available measurements were averaged to a daily value. Therefore, daily mean values shown in Figure 3 are means over the entire area of consideration and while the L4 IST product always has the same number of valid pixels used for the daily mean, the single-sensor products have a varying number of available measurements depending on data quality and cloud coverage. The L3S product is the combination of all single-sensor products so its average is based on all available points from all single-sensor products."

Regarding the differences in overpass times and differences in Figure 2, this is an example plot to demonstrate the transition from single-sensor products (upper 3 panels) to super-collated and optimally interpolated fields (lower 2 panels). Differences between the single sensor products shown in the upper 3 panels are associated to orbits, swaths, resolutions, instrument footprints, IR instrument characteristics, retrieval algorithms, etc. For the single day presented in Figure 2 differences are also associated with the number of available observations per

sensor/dataset, e.g. see white areas for AASTI and AATSR and the almost full coverage for MODIS. As an example, for AASTI and MODIS which rely on the AVHRR instrument on multiple platforms and the MODIS instrument on the Acqua and Terra platforms, there are more than 10 overpasses during for a given date. To the contrary, AATSR on Envisat had a revisit time of multiple days so full coverage of the GIS was very limited.

Please see clarification in Lines 169-172: "The coarser spatial resolution of AASTI (top left) compared to MODIS (top middle) is visible, resulting in AASTI grid points with missing information while MODIS daily aggregated L3 data offer superior coverage over the GIS. The sampling of AATSR (top right) with its narrow swath and lower temporal resolution results in characteristic artefacts resembling the ENVISAT platform orbit. Such artefacts do not appear nor for the MODIS neither the AASTI products."

- *Figs 5-6: biases are systematically negative as LST (observed mostly during day) is compared to a daily mean (including night). Therefore, both product are not comparable for me because the passing time of LST is not representative of the daily amplitude of observed temperatures (This issue is moreover mentioned in the conclusion). The passing time should be considered to have a fair comparison or at least, only the day hours should be used to compute the PROMICE average.*

The satellite observations are based on infra-red instruments and not optical, which would result in a day/night bias. Studies have shown seasonal cloud cover dependence over Greenland but not a daily one, see Nielsen-Englyst et al. 2019. In Figure 5, mean values from the single-sensor products, L3S and L4 are compared against mean values from the PROMICE stations. PROMICE mean values are systematically higher, hence the negative biases. As already mentioned in the manuscript, Lines 126-133, the main difference between the IR datasets and PROMICE is the clear-sky vs all-sky conditions, as in the former case colder temperatures are experienced compared to the latter: "The satellite products used in this study represent the clear-sky IST as the IR satellite sensors cannot observe the surface through clouds. As a result, a clear-sky bias is usually observed when comparing averaged clear-sky surface temperatures against averaged all-sky temperatures (Koenig et al., 2010; Comiso et al., 2003). Nielsen-Englyst et al. (2019) used

PROMICE observations to estimate the clear-sky bias intro­duced when averaging using different temporal windows. Us­ing a 72-hour averaging window, they found a clear-sky bias of -0.96°C when PROMICE stations located in the middle/upper ablation zone and the accumulation zone were used. Here, the clear-sky bias of 0.96°C has been added to the satellite prod­ucts in order to provide an estimate of the corresponding all-sky daily IST fields, which can be compared to the all-sky ISTs observed by PROMICE and IceBridge."

- *Section 4.2: the mean LST from satellite products over 2012 should be compared in 2D by considering all the available data and by considering only the pixels present in each data set. The differences between the products are due to the passing time? the cloud mask which is different in each data set? or for the same area at the same time, observed LST is different? More­over, the amount of missing data (still in 2D) in each data set should be compared instead of showing the number of aggre­gated observations in Fig 10.*

The number of aggregated observations in Figure 10 aim to demonstrate the number of available observations used to gen­erate the L4 product and to compute the monthly means shown in Figure 9. Nonetheless, we have removed it and included the information in the text, which was also a request from reviewer 2.

We have now estimated mean IST from MODIS, AASTI and L3S over 2012 along with the associated number of obs used for each dataset. This is now figure 5 and the associated text in Lines 197-202. "The spatial variability of mean annual IST val­ues over the Greenland Ice Sheet for 2012 from AASTI, MODIS, the L3S and L4 OI products are shown in Figure 5 along with the number of observations used to derive the means. For the MODIS and AASTI datasets, the intermediate L3 gridded products were used for the estimates, i.e. the original 1-km and 4-km L2 observations were re-gridded to the 5-km final grid. MODIS mean IST was significantly lower over the en­tire Greenland Ice Sheet compared to the AASTI estimates, al­though significantly more observations were available for the former. Mean IST from the L3S and L4 OI products appear more similar to AASTI mean estimates."

**3   About the assimilation:**

- *The passing time needs to be take into account for me. It is particularly relevant in May when melt occurs sometime only at local noon. By assimilating a daily average, this smooths the daily amplitude in the energy balance model and then the production of melt. It is particularly relevant at the beginning and at the end of the melting season.*

  The L4 IST dataset would ideally be produced in at least 6 hourly timesteps for the purposes of assimilation into our SMB scheme. Unfortunately, this is not possible within the current study and with the limitations of available data, which is why we have focused on using the daily means which have already been produced as an initial test of the system using the L2 data from the ESA CCI LST phase 1. The reviewer is quite correct that this introduced smoothing affects the energy balance calculations and therefore biases the derived SMB. Improving this scheme is part of planned future work. We have made this more clear by adding a few sentences in the text describing the assimilation in the methods section, see lines 158-163: "The L4 IST product is available once daily, yet modelled IST is dependent on the full surface energy balance and therefore highly variable in space and time, assimilating the L4 IST product inevitably introduces some biases. The aim of this study is therefore to act as a proof of concept for the assimilation of satellite derived data into the model. It is also for this reason that we focus on the month of May, a period with highly variable IST and surface melt where the addition of satellite observations is likely to give the most added value in identifying surfaces close to the melting point."

- *During the peak melting season, melt occurs every time and as IST is limited to 0°C, assimilating of not IST does not change the SMB results explaining why the focus has been made here over May.*

  The reviewer is correct that as melting surfaces are forced to be 0°C, there is no change in SMB over surfaces where both model and IST data suggest it is melting. However, it is very rare for the entire ice sheet to melt, so there can in fact be changes in the positive direction if the assimilated IST gives a colder surface than the model during the melt season. In May, the ice sheet typically goes through a period of rapid change

as melt starts. We have added this, see lines 158-163 and the reply above.

- *While the aim is to improve SMB and surface melt, nothing is said about the differences in the cumulated melt amount, runoff, SMB, ... between the control and assimilation run. Only IST is compared between both simulations. But, in term of SMB and melt, it is not clear if this improves or not the results. As the temperature is forced in the SMB model, it is not very relevant for me to evaluate IST only. The melt extent should be compared with a microwave derived product for example to see the interest here of assimilating a daily product. Moreover, integrated over the whole season, what is the impact on the production of melt, runoff and refreezing?*

Response: The aim of this paper is to evaluate a remote sensing product and show examples of how can it be utilised to analyze the IST over Greenland and to interpret regional climate and surface mass balance models of the Greenland ice sheet. Therefore, as the focus is not on the melt and the manuscript is already extended, according to comments from reviewer 2, we refrain from adding new analyses/figures. Please also see clarification in the Discussion, lines 330-334.

Future work will focus on this aspect and in refining the assimilation scheme to take into account some of the mentioned features. Nonetheless, we have done a quick comparison with the melt area between the control and the assimilation run compared with the melt area derived from passive microwave observations and published by NSIDC in order to assess the modelled melt area (see figure below). The assimilated IST run shows a closer match with estimates of melt area in western Greenland as well as in the NE and SE, but areas around the NW of Greenland are better represented in the control run. The extra analysis required to understand the biases and implications of the current rather simple assimilation scheme is beyond the scope of this paper but will be included in future work.

[Figure]

Figure 2: Quick comparison of melt area between control and IST assimilation models with NSIDC melt area derived from passive microwave.

---

## Author Comment (AC2)

**Response to comments from Referee #2 for TC-2021-384**

**1  General comments**

- *The paper is fundamentally a comparison of several ice surface temperature products over Greenland for the calendar year 2012, assessing their relative performance against ground-based AWS (PROMICE stations) and airborne radiometers (IceBridge KT-19 profiles) for that year. The study emphasizes a new Level-4 (optimally interpolated, gap-filled, gridded) data set, describing how it is produced, compares it with several other products and validation data sets, and then uses the L4 data as an input for an SMB model to determine its effect on SMB estimates for 2012, a year with record surface melt and run-off. The paper is a bit confusing to read. The title needs to be changed because it gives the reader the impression that it will be a data-set focused paper, on the new product specifically, over an extended period. A better title might be: "Multi-sensor assessment of Ice Surface Temperature products for Greenland's 2012 melt season". And then introduce the new merged product within the Introduction. But I think that a better approach would be to convert this paper into an ESSD paper, and then write a shorter paper focused on the application of the data set to the 2012 melt season.*

  We thank the reviewer for the constructive suggestion and we have partially modified the title. However, as the paper is presenting the dataset for the full 2012 year and not only the melt season, we refrain from adding that part. Furthermore, as this is a demonstration study to show what a L4 IST product can be used for, given that it can be produced for more years, we also refrain from specifying only 2012 in the title. This has now been clarified in the 1st paragraph of the discussion, lines 330-334: "In this study, infra-red observations from the reprocessed archive of the ESA LST_cci project and

the AASTI dataset were utilised to demonstrate the capability for generating a Level 4 Ice Surface Temperature product over the Greenland Ice Sheet, based on existing long-term, homogenized datasets from space-bourne sensors. The aim was to demonstrate the generation, quality and performance of the new L4 IST product compared to its single-sensor predecessors and in situ observations and finally the applicability of such a product for monitoring IST over Greenland and its utilization in a Surface Mass Budget model."

Regarding separating the manuscript, please kindly note that we refrain from doing so as manuscripts of this format are typical when introducing multi-sensor products in order to demonstrate their applicability, e.g. Høyer and Karagali, (2016). This format offers an overview of the datasets used, their performance, the new product and its performance, along with what can it be used for thus helping the readers understand how multi-sensor products are generated and how can they be utilised.

- *This paper seems to be trying to do several things at once: describe a new data product, validate it, discuss its benefits / limitations, investigate the annual cycle for the Greenland ice sheet in 2012, and finally the potential advantage of an L4 IST for SMB analysis. It would be far easier to follow the research if first there were a paper on the L4 data set for the full time-period it can cover, with multi-year validation and something like a climatology for the ice sheet – and then a study of the 2012 melt season and SMB models using it.*

    Please see the response to the previous comment and the discussion in Lines 396-401: "The L4 OI IST dataset generated and presented here was the result of a user case from the ESA LST_cci project and was only generated for the selected year of 2012 to assess the impact and applicability of such a product over the Greenland Ice Sheet. Ideally such a product can be expanded to cover the entire period of available L2 input data, thus resulting in more climatologically relevant time scales in the order of 20 to 30 years. Such a task will become significantly more relevant during the second phase of ESA LST_cci during which the current suite of products will be improved and temporally extended and new products will be included, e.g. the AVHRSS series (NOAA 7-19 and MetOp-A/B/C)."

- *I think the paper could be close to publishable, but as an ESSD paper. The revised title suggested above would lead to a shorter,*

*application-focused, tighter paper that would not do justice to introducing the new data set and its usefulness. The major revisions required are a re-write, fairly comprehensive, to make it more focused on this 'data product' target, and to describe the full multi-year time-series that can be derived for the L4 product. A separate paper could then be developed, if desired, on the unusual climate aspects of 2012 as revealed in the IST all-sky result in Greenland, with a comparison in more detail with the existing literature on the 2012 summer there. As it stands, the manuscript seems to wander between describing a small piece of a potentially important data product (the L4 IST) and some kind of analysis of the geographic distribution of unusual temperatures in 2012. The ESSD paper would re-focus on introducing the study more clearly, and perhaps revising some of the graphics, and reducing the number of graphics (finding other ways to show the validation/ comparison information). I leave it to the editor to decide, of course, but I think the clearest path is to use most of this work for an ESSD paper, and then submit a shorter paper on analysis of 2012 to The Cryosphere. Sorry, it probably shows in this writing that my thinking on the text evolved over the couple of afternoons I reviewed it.*

Please see response to previous comment about separating the manuscript.

**2  Detailed comments**

*Many comments are embedded in the annotated .pdf of the paper, submitted with this review.*

We have addressed them individually, please see below.

- *change to 'Level 4' for the title*
  Corrected.

- *Abstract is much longer than it needs to be, rambling. 450+ words, could easily be 250.*
  We have shortened the abstract.

- *Line 16: 'upstream' not needed.*
  Removed.

- *Line 19: please give the total range of melt days – from 90 to 2, with the greatest e.g. in the southwest, and the area north of Summit with 0 to 1.*

  The sentence refers to "almost the entire GIS" for which the range of melt-days is actually 1-5. Some parts of it - far from almost the entire GIS though - experienced high number of melt-days and this description is already mentioned in the text.

- *Line 35: 'survive as liquid' - do you mean run-off? perennial subsurface firn aquifers? this is unclear and confusing.*

  Here we are referring to irreducible liquid water that fills up the pore space in the snow pack. In some parts of Greenland this also leads to the formation of perennial firn aquifers when local conitions allow. We have updated the sentence, see Lines 30-33: "Also important are processes of meltwater percolation into the snow and firn (snow that has survived at least one annual cycle) where meltwater can be retained as a liquid if there is sufficient pore space and may refreeze if the cold content is sufficient, potentially forming aerially extensive ice layers (Broeke et al., 2009; Ettema et al., 2010; Machguth et al., 2016; Reijmer et al., 2012)."

- *Line 36: author name is 'van den Broeke'.*

  Corrected.

- *Line 65: change space-bourne to 'satellite'.*

  Changed.

- *Table 1: 'swath' is used incorrectly - do you want to indicate the sensor swath width? then add a separate column, 'swath'. Swath widths for these satellites are in the 100s to 1000s km.*

  Swath is meant to inform about the format of the data, as L2 products are in the original satellite swath format and not gridded to a regular lat/lon grid. Understanding it may cause confusion, it has been removed.

- *Line 97: no hyphen needed up-welling.*

  Corrected.

- *Line 106: how wide is the kt-19 swath?*

  The instrument footprint is 15 m, and this information is now included in the text, see Line 103: "Due to the high resolution footprint of the KT-19 instrument - approximately 15m at 450

above ground (Studinger, 2020) - which results in high variability of the observed radiometric surface temperature, IceBridge observations were averaged for every kilometre to make them more comparable to the lower resolution satellite data."

- *Line 113: is the word 'upstream' necessary here?*

  Removed.

- *Line 132: what is the error on the comparison with PROMICE stations? The simple bias correction, adding the regional long-term offset to derive all-sky IST is concerning.... different elevations are likely to have different clear-sky / all-sky biases.*

  This is specifically analysed in the manuscript Nielsen-Englyst 2019, as also mentioned in the text, for different parts of the Greenland Ice Sheet. In the present manuscript, we only use PROMICE stations of the upper ablation zone and accumulation zone were used to ensure comparisons with the satellite IST were performed only over permanently snow/ice covered surfaces. Therefore the bias correction used, as derived in Nielsen-Englyst 2019, is the one for upper ablation and accumulation zones. This is all explained in the text in Lines 94-95: "Only PROMICE data from the upper ablation and accumulation zones were used to ensure that data are only acquired over permanently snow- or ice- covered surfaces."

  and in lines 126-133: "The satellite products used in this study represent the clear-sky IST as the IR satellite sensors cannot observe the surface through clouds. As a result, a clear-sky bias is usually observed when comparing averaged clear-sky surface temperatures against averaged all-sky temperatures (Koenig et al., 2010; Comiso et al., 2003). Nielsen-Englyst et al. (2019) used PROMICE observations to estimate the clear-sky bias introduced when averaging using different temporal windows. Using a 72-hour averaging window, they found a clear-sky bias of -0.96°C when PROMICE stations located in the middle/upper ablation zone and the accumulation zone were used. Here, the clear-sky bias of 0.96°C has been added to the satellite products in order to provide an estimate of the corresponding all-sky daily IST fields, which can be compared to the all-sky ISTs observed by PROMICE and IceBridge."

- *Line 145-146: seems like there is a grammar problem in this sentence.*

  Corrected.

- *Line 150: straight away a bit colloquial, use 'immediately'.*

  Corrected.

- *Line 170: colder IST change to 'lower' - temperatures are high or low, not warm or cold.*

  Corrected.

- *Line 176: why? just eliminate April for this data set.*

  Corrected and removed, see updated figure 4 and line 184-185: "AATSR was only available until the beginning of April, thus no monthly value was calculated."

- *Table 3: change this table note to read: Winter mean temperatures were determined by averaging January, February and December of 2012.*

  Modified.

- *Line 191: Please describe the problem with the cloud mask - e.g., it does not eliminate cirrus cloud well enough?*

  Please see description in Lines 203-207: "The primary reason for the lower LST_cci v1.0 MODIS and AATSR IST values, used in the present study, is the type of cloud masking applied in the first version of the data. No post-filtering or implementation of the cloud masking techniques (later developed within the LST_cci for both instruments) were applied in the v1.0 of the data presented here but only the standard operational cloud mask; this frequently failed to properly flag clouds, which are typically colder, resulting in lower surface temperature values."

- *Figure 2 – why is this a wintertime assessment when this study is about a melt season excerpt of the product? Would not an April 2012 comparison be more appropriate?*

  The study is not about the melt season, as the product is made, validated and analysed for the entire year of 2012. The reason 2012 was selected is because a significant melt event occurred and this is clarified in the abstract, introduction and discussion. Figure 2 aims to demonstrate the added value of the L4 product compared to single-sensor datasets in terms of spatial coverage. The wintertime example is relevant as cloud cover, impervious to IR radiation, is higher. This is also explained in the text, see line 166: "The L3 products are aggregated for January 9, 2012 - winter time when cloud cover, impervious to IR radiation, is higher - into the L3S product (bottom left) ..."

- *Figure 4: widen this graphic so that you can make it more clear that you are clustering the monthly means with slight offsets for the different data sets. Remove April for AASTR since it is a partial month.*

  Done.

- *Figure 6: perhaps use a table for this presentation.*

  We have removed this figure and added a table instead.

- *Figure 7: I don't see the value in this kind of detail – would it not be better to simply describe the overall bias for each IST data type relative to the 2012 IceBridge flights? Another graphical, map-based way to do this would be 4 outline maps of Greenland, one for each IST data type, with the flight tracks shown, colored along the track by offset (difference between IST and KT-19) smoothed to, e.g. 10km, on each track. Really clever addition would be to show the s.d. for the 10km as a grey width to the colored line. That, and a table summarizing the whole-season 28-flight average bias and offset.*

  Due to the amount of available flight tracks (see new Figure 1, with map of Greenland and all IceBridge flights), plotting all 28 in one figure per dataset, along with the standard deviation as a shaded area around each flight path results to an incomprehensible figure due to the overlaps. Therefore, we refrain from adding such a figure. Furthermore, the graphical representation of the biases per flight campaign and product, as described in Lines 243-245 serves to demonstrate the variability from campaign to campaign which is, e.g., significantly more pronounced for MODIS, see: "MODIS was cold compared to the flight measurements, manifested as a negative bias (–5.19°C±4.8°C), and with a pronounced variability during the period evident from the oscillating bias (from –14.15°C to 2.20°C) and standard deviation values (from 2°C to 7.2°C)."

- *Figure 8: This might be merged with a condensed version of Figure 7, as I suggested in my note for Fig7.*

  Please see response above and modified Figure 8, where the map with the single IceBridge Flight has been omitted and all IceBridge flights have been presented in the modified Figure 1.

- *Figure 10: This graphic would only be of use in a data description paper; its not really useful in a Cryosphere paper.*

  The figure aimed to provide information about the number of observations available to generate the L3S and L4 products

which is then used to derive the mean values shown in Figure 9. Nonetheless, it has now been removed and its information conveyed through text.

- *Figure 11: the left graphic might be better as an addition to a re-shaped fig9; the center and right graphics here are a nice outcome of the L4 product, but are more appropriate for an analysis of the 2012 melt season in comparison with other melt-day product. On this point, the color bar for the right graphic should be revised to a different palette, and adjusted to show the 0 to 50 day range more clearly. It would appear that the total number of melt days is low relative to other measures of 2012's melt season – something to evaluate in your 2012 analysis paper.*

As mentioned above, since we refrain from separating the manuscript in two, the figure remains as is originally. Nonetheless, we have modified the color schemes to different palettes so now the right panel has the max number of available melt days maintained and an intense color transition at 50 days so the 0-50 days is more visible.

- *Figure 12: not terribly important, but the projection is rotated several degrees ccw here.*

Corrected (now figure 11, since a figure has been omitted as per a previous comment).

- *Figure 13: I think this is better presented in a map view so that readers can incorporate geography into their assessment of the results.*

This figure was complementary to figure 6, which has been turned into a table according to the reviewer's request and thus, also to align with the comment about many figures, we also modified this figure to a table. Please do keep in mind that the geographical information of where the PROMICE stations are located is conveyed in Figure 1 of the manuscript (left panel).

- *Figure 14: I just don't think this graphic is informative without a lot of work on the reader's part to examine the flight paths, presence of clouds, weather that day... is there more information for the analysis than the bias and SD reported at the lower right of the graphs?*

This figure is presented in a similar way as for the comparisons of the satellite data with the IceBrige campaigns (Figure 7). It can provide useful information on the variability of the campaigns and the performance of the two types of simulations,

therefore to maintain the consistency with the presentation of validation results we refrain from removing it. Please also refer to response about Figure 7.

- *Figure 15: Perhaps you could pick 3 or 4 flights with a story to tell, and show those, with the map at the right side. Lots of white space in this.*

  We have modified this figure (now figure 13) to be in alignment with Figure 8.

- *In general, too may figures of low value in the information and 'story' they convey.*

  We have converted some of the figures to tables and have removed some, yet we have included some new due to the request of Reviewer 1, thus the total number is now reduced to 13 from 15 in the original version.

- *Line 321: 2012 in reference?*

  It is actually Table 1 of the 2013 paper, as is already mentioned in the text.

- *Line 339: did you mean to say 'not very similar' here? otherwise there is a logical problem with the next sentence.*

  The spectral response functions (SRF) of the KT-19 instrument are similar to the satellite IR SRF, nonetheless its footprint is not, thus the next sentence "Therefore, the results from the inter-comparison should not be viewed as an estimate of the uncertainty of the satellite products." is actually valid. The text has now slightly been modified, please see lines 352-355: "In addition, although the Spectral Response Functions (SRFs) of the IceBridge KT19 instrument are very similar to the actual IR satellite SRFs, the instrument footprints are different. Therefore, the results from the inter-comparison should not be viewed as an estimate of the uncertainty of the satellite products."

- *Line 355-356: add what ?Hall et al., 2013? reported as the average summer MODIS IST for Greenland (multi-year).*

  This has been modified for clarity, see lines 374-376: "Hall et al. (2013) reported more than two melt days for most of the GIS during the melt season of 2012, based on MODIS data, which also indicated the warmest summer in the MODIS record with mean IST of −6.38±3.98°C, in good agreement with the mean summer IST from the L4 IST product reported in Table 3."

- *Its a good paper, well-written, but its trying to surf the boundary between a data paper and a science study.*

  We very much thank the reviewer for the time used to read this manuscript and for the constructive comments. We have tried to clarify this point in the discussion, see Lines 330-334:"In this study, infra-red observations from the reprocessed archive of the ESA LST_cci project and the AASTI dataset were utilised to demonstrate the capability for generating a Level 4 Ice Surface Temperature product over the Greenland Ice Sheet, based on existing long-term, homogenized datasets from satellite sensors. The aim was to demonstrate the generation, quality and performance of the new L4 IST product compared to its single-sensor predecessors and in situ observations and finally the applicability of such a product for monitoring IST over Greenland and its utilization in a Surface Mass Balance model."

  and Lines 396-401: "The L4 OI IST dataset generated and presented here was the result of a user case from the ESA LST_cci project and was only generated for the selected year of 2012 to assess the impact and applicability of such a product over the Greenland Ice Sheet. Ideally such a product can be expanded to cover the entire period of available L2 input data, thus resulting in more climatologically relevant time scales in the order of 20 to 30 years. Such a task will become significantly more relevant during the second phase of ESA LST_cci during which the current suite of products will be improved and temporally extended and new products will be included, e.g. the AVHRSS series (NOAA 7-19 and MetOp-A/B/C). "

---

## Author Response (AR2)

**Response to comments from Referee #1 for TC-2021-384**

- *OK to keep the manuscript mainly as it has been and to leave the melt/SMB evaluation for a further paper. But as only temperature is compared after assimilation of IST without any comparison/evaluation of SMB, using the word "surface energy balance model" instead of "SMB model" or "snow model" through the paper would be more fair. If the authors prefer, ok to keep "SMB" model (if it is the model name) but in this case, it is important to well specify that only IST is discussed/evaluated here and not SMB/melt.*

  We have opted for the second option, i.e. to maintain the SMB title and clarify that the results/assessment only cover the IST component in this manuscript. Please see examples in lines 49-50, 79-80, 161-164, 305-308, 309-310, 316, 322, 327-330, 334-338, 402-405, 442-443, 445-450, 452-453.

- *For example, the legend of Fig 12 should be "snow model control run" vs "snow model assimilation run" instead of SMB.*

  We have not modified the titles of the two panels in Figure 12 yet according to the response above we have clarified that only simulated IST is evaluated and not SMB and melt.

- *Idem, in the abstract, the sentence: "inclusion of the L4 IST dataset improved the model performance during the key onset of the melt season, where model biases are typically large" is too vague and needs to be precise to something like "inclusion of the L4 IST dataset improved the modeled IST during the key onset of the melt season, where model biases are typically large and could impact amount of simulated melt".*

  Sentence was modified accordingly.